# Comprehensive Assessment of Somatostatin Receptors in Various Neoplasms: A Systematic Review

**DOI:** 10.3390/pharmaceutics14071394

**Published:** 2022-07-01

**Authors:** Shista Priyadarshini, Derek B. Allison, Aman Chauhan

**Affiliations:** 1Internal Medicine, Guthrie Robert Packer Hospital, Sayre, PA 18840, USA; shista.sp@gmail.com; 2Department of Pathology & Laboratory Medicine, University of Kentucky, Lexington, KY 40536, USA; derek.allison@uky.edu; 3Markey Cancer Center, University of Kentucky, Lexington, KY 40536, USA; 4Division of Medical Oncology, University of Kentucky, Lexington, KY 40536, USA

**Keywords:** somatostatin receptor, SSTR, somatostatin receptor prevalence

## Abstract

Somatostatin receptors (SSTR) are expressed in various neoplasms and can be targeted for both diagnostics as well as therapeutics. This systematic review aims to compile and discuss the prevalence of somatostatin receptor expression in various neoplasms. We performed a literature search from Google Scholar and PubMed using relevant keywords to look for all publicly available data regarding SSTR expression in various cancers. Both histopathological and radiographical studies were included for SSTR assessment. We found that many cancers express SSTR with varying prevalence. SSTR is now a well-established theranostics biomarker. We now have highly sensitive and specific diagnostic modalities like gallium 68 DOTATATE and copper 64 DOTATATE scans to screen for SSTR-2 and then target it therapeutically with lutetium 177 DOTATATE. A thorough understanding of SSTR expression in other tumors will open the channels for exploring potential SSTR targeting.

## 1. Introduction

The somatostatin receptors (SSTR), well-known for their association with well-differentiated gastroenteropancreatic neuroendocrine tumors, have recently been in the limelight due to the resurgence of theranostics. SSTR was discovered and studied by Brazeau et al. in 1973 because of its inhibitory role on growth hormone secretion from pituitary cells [1]. Belonging to the G-protein-coupled heptahelical receptor superfamily [2], the somatostatin receptor is a cyclic transmembrane tetra-decapeptide (14 amino acid) protein with regulatory and secretory functions (shown in Figure 1). There are primarily 6 subtypes: SSTR1, SSTR2A, SSTR2B, SSTR3, SSTR4, and SSTR5 [3]. Widely expressed throughout the body, these receptors serve many physiological functions depending on the cell type and have anti-proliferative effects. These are found not only in normal tissues but also in various pathological conditions such as chronic inflammatory disorders and many cancers. Among cancers, these have been widely used in diagnostic and treatment modalities in well-differentiated gastroenteropancreatic neuroendocrine tumors (GEPNET); however, very little is known about receptor expression in various non-GEPNET neoplasms. This manuscript focuses on the expression of somatostatin receptors in non-GEPNET neoplasms.

Over the past two decades, the interplay between somatostatin analog with somatostatin receptors and its ensuing intracellular processes has led to a new era in the field of NET diagnostics and therapeutics. The conventional approach was to inject octreotide, a somatostatin analog, linked to a radioactive isotope indium-111, which binds to cells expressing somatostatin receptors, is internalized and is traced on a SPECT (single-photon emission computed tomography) nuclear scan. DOTATATE [4], an 8-amino acid polypeptide SSTR agonist, was found to have better sensitivity and specificity. With continuing advancements, gallium-68 replaced indium-111, and Ga^68^-DOTATATE emerged as a novel invention, which shows far superior nuclear uptake and spatial resolution on PET/CT compared to indium-111 on SPECT [5]. Improvement in diagnostics paved the way for therapeutics. Lutetium-177(a short-range beta emitter) was linked to DOTATATE, which allowed internalizing of radioactivity to desired cells and proved to be a significant advancement in therapy for NETs. In 2018, Lu^177^ DOTATATE was the first somatostatin-targeted radionuclide-based therapy approved by the US FDA for metastatic progressive GEPNETs [6]. 

This recent success in NETs has led us to explore the potential to utilize the same principle in the diagnosis and targeted therapy of various non-GEPNET tumors. Currently, data on somatostatin receptors and their expression in various non-gastroenteropancreatic neuroendocrine tumors is limited. In this review, we have collated what is known about the differential expression of somatostatin receptors in these tumors. 

## 2. Materials and Methods

This systematic review is based on a thorough assessment of the published literature regarding the expression of somatostatin receptors in various neoplasms other than GEPNETs. An extensive review of the literature was conducted using Google Scholar and PubMed. The following keywords were used: “somatostatin receptors prevalence”, “somatostatin receptors prevalence in malignancies except well differentiated gastroenteropancreatic neuroendocrine tumors”, and “somatostatin receptors prevalence in specific malignancies”. Articles pertaining to SSTR expression in Merkel cell carcinoma, neuroblastoma, small and large cell neuroendocrine carcinomas and paraganglioma/pheochromocytoma were allowed. The search was further broadened using references cited in relevant identified literature. Studies describing SSTR expression in human patients and tumors were selected for final assessment. Non-English language manuscripts were excluded. 

We used the PRISMA guidelines to collect data for this manuscript. With the predefined keyword searches, we identified approximately 150 manuscripts. Because of limited access to a subset of articles, 123 articles were selected for further analysis. Among these, the majority focused on GEPNETs. After excluding articles solely focused on GEPNETs, (Merkel cell carcinoma and para/pheos were included), 57 articles remained. Of these, three articles included animal studies, which were excluded. We finally collected the data and summarized our findings based on 54 published articles. The PRISMA flow diagram is displayed in Figure 2.

The following tumors showed SSTR expression: lymphoma, gastrointestinal carcinomas, head and neck carcinoma, lung carcinoma, Merkel cell carcinoma, malignant melanoma, meningioma, neuroblastoma, thyroid carcinoma, thymoma, breast carcinoma, pheochromocytoma, paraganglioma, prostate carcinoma, ovarian carcinoma and endometrial carcinoma.

## 3. Results and Discussion

### 3.1. Lymphoma

Lymphomas are a heterogeneous group of hematogenous cancers which range from indolent to aggressive. While some lymphomas can be effectively treated, many subtypes are still in need of good treatment strategies, especially for relapsed refractory disease. We analyzed the available literature to explore the possibility of targeting SSTR as a potential therapy option. Briefly, Dalm et al. [7] investigated 10 patients (including 6 orbital lymphomas, 2 with Hodgkin, and 2 with non-Hodgkin lymphoma) in 2004. Quantitative RT-PCR, autoradiography using I^125^Tyr^3^octreotide, and immunohistochemistry (IHC) were used to study the expression of SSTRs. Low transcript expression of SSTR 2 and 3 was noted by quantitative PCR. Likewise, autoradiography showed low binding, and immunoreactivity was undetectable by IHC. A year later, Ferone and colleagues [8] stated that the uptake of ^111^In-DTPA-octreotide was lower in lymphomas than in NETs. The sensitivity of somatostatin receptor by scintigraphy was found to be variable, in the range of 95–100% in Hodgkin lymphoma and 80% in non-Hodgkin lymphoma (NHL), but showed little utility as a specific diagnostic tool for lymphomas. However, they stated that as an exception, somatostatin receptor scintigraphy (SRS) could be used in extragastric MALT-type lymphoma for staging. In congruence with this, a clinical trial from Austria [9] studied the role of SSTR expression in extragastric MALT-type lymphoma. The study included 30 patients, of which 24 had primary extragastric MALT-type lymphoma, 5 with initial gastric dissemination to extragastric sites, and 1 with primary parotid MALT-type lymphoma spreading to the stomach, lung, and lymph nodes. SSTR scintigraphy with ^111^In-DTPA-octreotide showed positive scans in all primary extragastric lymphomas and negative scans in lymphomas with gastric origin disseminating to other sites. The octreotide scans were also used to assess for resolution or persistence of disease and thus the effectiveness of therapy, which was confirmed with MRI and biopsy. The trial concluded that SSTR expression could be used to differentiate extragastric from gastric MALT-type lymphomas and could be used for staging and monitoring disease progression. In contrast, another study from Germany [10] compared CXCR4, a chemokine receptor, and SSTR expression in MALT-type lymphomas of gastric and extragastric origin by immunohistochemistry using rabbit monoclonal antibodies against SSTR. Among 55 cases studied, CXCR4 was seen in 92% of cases, while SSTR subtypes were less frequent and as follows: SSTR5 (50%), SSTR3 (35%), SSTR2A (27%), SSTR4 (18%), and SSTR1 (2%). When compared to the prior study, SSTR 3, 4, and 5 expression was noted to be higher in gastric versus extragastric MALT-type lymphomas. Interestingly, SSTR5 negativity in these extragastric tumors was associated with a poor prognosis. Another pilot study from Finland [11] used FDG PET/CT and ^68^Ga-DOTATATE PET/CT to assess SSTR 2, 3, and 5 expression in 21 newly diagnosed lymphoma patients. The Krenning score is a method to assess the tracer uptake by neuroendocrine tumors on an octreotide scan. A Krenning score ≥ 2 on uptake of ^68^Ga-DOTANOC was considered positive. Although 20 patients were positive on FDG, only 13 (62%) were noted to have ^68^Ga-DOTANOC uptake. The highest Ga uptake was seen in nodular sclerosing Hodgkin lymphoma and diffuse large B cell lymphoma (DLBCL). In most subtypes, SSTR2 showed the strongest immunoreactivity, while SSTR3 and SSTR5 were most often negative. The study concluded that SSTR expression in some lymphoma subtypes could be targeted for treatment. In a comparative study from the Netherlands [12], SSTR scintigraphy with ^111^In-Pentetreotide was used prospectively in the initial staging of 50 untreated patients with low-grade NHL. The findings were compared with conventional staging methods. Somatostatin scintigraphy was positive in 84% (42/50 patients) of cases. In 20% (10/50) of cases, scintigraphy also showed new lesions. However, in 38% of cases, lesions visualized by conventional methods were missed on scintigraphy. Overall, scintigraphy was seen to have high specificity (98–100%), but sensitivity was low (62% in supra-diaphragmatic and 44% in infra-diaphragmatic lesions). With limited sensitivity, it was concluded that scintigraphy should not be used for work-up but rather for staging in select patients with low-grade NHL. We believe that SSTR expression holds diagnostic and therapeutic potential in some sub-sets of lymphomas; however, it is imperative to study SSTR expression with the current generation of SSTR imaging, which is highly sensitive and specific for the detection of SSTR.

### 3.2. Endometrial Neoplasm

Somatostatin receptors have been found to be expressed in endometrial tissues. A recent study in 2018 [13] used RT-PCR to evaluate SSTR mRNA transcript expression in patients with ectopic and eutopic endometrium and compared the expression to a control group. In endometriosis with ectopic endometrial tissue, the expression was as follows: SSTR5 (96.7%), SSTR4 (50%), SSTR3 (53.3%), SSTR2 (70%), and SSTR1 (43.3%). Among 12 patients studied with eutopic endometrium, similar results were seen, as follows: SSTR5 (83.3%), SSTR4 (58.3%), SSTR3 (58.3%), SSTR2 (41.7%), and SSTR1 (33.3%). In comparison, normal endometrium generally had a lower expression of SSTR except for SSTR5. 

### 3.3. Prostate Neoplasm

Prostate cancer is among the most common cancers afflicting males. Treatment of metastatic refractory disease can be challenging. In 2020, a German study [14] explored novel theranostic therapeutic targets. The study analyzed SSTR and CXCR4 expression in 276 prostate cancer samples and reported a very low-intensity expression of both SSTR and CXCR4. The expression was noted as follows: SSTR5 (10.5%), SSTR3 (0.7%), and SSTR2A (9.1%), while no expression of CXCR4 or SSTR1 was identified. This study concluded that somatostatin receptors had no therapeutic potential for prostate cancer.

Savelli et al. [15] studied six patients with castration-resistant prostate cancer (CRPC) for SSTR expression with PET-CT after ^68^Ga-DOTANOC administration. Among the six subjects, two patients were found to have increased DOTANOC uptake. Literature suggests ^68^Ga-DOTANOC lacks affinity for SSTR3, whereas ^68^Ga-DOTANOC had a high affinity for SSTR 2, 3, and 5. In 2014, another group of researchers from Germany [16] evaluated SSTR2 expression in prostate cancer via quantitative IHC. The study analyzed 3261 specimens. Normal prostate epithelium always showed membranous staining for SSTR2. Among the prostate cancer specimens, only 13% showed cytoplasmic and membranous moderate to strong staining. Overall, SSTR2 expression in the specimens was noted as follows: absent in 56.1%, weak in 31%, moderate in 8.5%, and strong staining in 4.4% of specimens. The SSTR staining was seen to have an inverse correlation to Gleason score and proliferative rates. In the negative-staining prostate cancers, there was a high likelihood of developing metastases. In this study, SSTR2 negativity predicted an unfavorable outcome and a poor prognosis. 

In 2000, Halmos et al. [17] studied mRNA transcript levels for SSTR1, SSTR2, and SSTR5 by RT-PCR in 80 prostate cancer specimens. SSTR detection was as follows: SSTR1 (86%), SSTR2 (14%), and SSTR5 (64%). A cyclic octapeptide somatostatin analog RC-160 was also used to study the affinity and specificity of binding sites. A total of 65% of these specimens showed binding with RC-160, and the affinity was similar in both high- and low-risk cancer states.

### 3.4. Gastrointestinal Cancers 

Somatostatin receptor expression has also been studied in gastric cancer. 

In 2012, a group of researchers from Italy [18] performed IHC for SSTR2A and HER2 to correlate expression with grading, staging, differentiation, and outcomes of gastric cancer. A total of 51 patients were studied, and SSTR2A expression was identified in 38 cases (74.5%). Interestingly, the prevalence was higher in well-differentiated (96%) and intestinal-type cancer (97%) compared to poorly differentiated (52%) and diffuse-type cancer (20%). Another study by Hu et al. compared SSTR3 [19] expression in 40 cases with gastric adenocarcinoma and 40 cases with normal gastric mucosa. Immunofluorescence, RT-PCR, and Western blot methods were used to analyze SSTR3 expression or mRNA transcript levels. SSTR3 expression was found to be higher in the normal gastric mucosa (62.5%) compared to gastric cancer (25%). Octreotide was found to have an inhibitory and apoptotic effect on gastric cancer and normal mucosa with SSTR3 expression. 

A preclinical study from China [20] evaluated the impact of the SST gene on the development of gastric cancer. The study was aimed at knocking out the SST gene from gastric cancer cells. The transformed cells had increased expression of SEMA5A protein and thus increased invasiveness and migration potential. In this study, SST gene expression was associated with decreased invasive potential, and its absence in gastric cancer cells led to increased metastatic potential.

Another study from 2002 [21] studied SSTR2 transcript levels and radiographic receptor expression in 100 cases of colorectal cancer and compared them to their corresponding normal mucosa by quantitative RT-PCR and ^111^In-Pentetreotide. Though SSTR2 was expressed in all tumors, the normal mucosa in these cases reported higher expression, though not of statistical significance. When compared to normal tissue, SSTR2 expression in colorectal cancer cells was inversely related to elevated CEA cells. This inverse relation of SSTR2 can be used as a poor prognostic marker.

While the above studies showed variable SSTR relation with gastric and colorectal cancers, similar animal knockout studies showed surprising results. A study from 2013 [22] highlighted the tumor-suppressive role of SSTR1 (in its methylated form) expression on gastric cancer cells. Prior to the intervention, promoter hypermethylation or SSTR1 downregulation was noted in EBV-positive gastric cancer but not in EBV-negative gastric cancer cell lines. Later, DNA demethylating agents were used, and SSTR1 knock-down was done using RNA interference, resulting in an increased proliferative tendency in the EBV-positive gastric cancer cells. SSTR1 knock-down gastric cancer cells also showed increased migration and invasiveness properties. The same study also analyzed the effect of SSTR1 transfection in nude mice with gastric cancer cells, and significant tumor shrinkage was noted when compared with the control mice. Therefore, inhibiting SSTR1 may not be an effective strategy for EBV-driven tumors. However, EBV-driven gastric carcinomas are extremely rare, and additional studies are needed in patient-derived xenograft models and human clinical trials.

### 3.5. Thyroid Carcinoma 

SSTR expression has been studied in thyroid and thymic cancers. One study analyzed SSTR mRNA transcripts in thyroid carcinoma monolayer cultures using RT-PCR [23]. Five thyroid cell lines, including papillary carcinoma, follicular carcinoma, anaplastic carcinoma, and five samples of normal thyroid tissue, were compared. In normal thyroid samples, expression of SSTR3 with variable staining was 100%, SSTR5 was strongly positive in 60% of samples. Furthermore, expression was weakly positive in 60%, 20%, and 0% for SSTR2, SSTR1, and SSTR4, respectively. Among follicular adenoma and carcinoma, expression of all SSTR subtypes except SSTR4 were observed, of which SSTR1 and SSTR3 were dominant. Papillary carcinoma had low expression via gel electrophoresis. In anaplastic carcinoma, SSTR showed variable expression of SSTR 1, 3, and 5 and faint expression of SSTR2. Few anaplastic carcinoma cell lines also expressed SSTR4. It was also noted that SSTR expression in monolayer cultures was greater when compared to xenografts. Human studies evaluating SSTR expression by either immunohistochemistry or radiolabeled somatostatin analog-based PET scanning are required for further characterization of thyroid cancers. 

### 3.6. Thymoma

In 2000, Ferone et al., from the Netherlands [24], studied the in vivo and in vitro expression of somatostatin (SS) binding sites in thymoma. ^111^In-DTPA-D-octreotide was injected, and uptake was noted in patients with thymoma with SPECT. RT-PCR was used to study SSTR mRNA transcripts. The samples were obtained from 1 thymoma patient and 3 normal thymi. The normal thymus was positive for SSTR1, 2A, and 3. No expression of SS mRNA was noted in the thymoma. In vivo thymoma tumor tissue showed SSTR1, SSTR2A and SSTR3 mRNA expression. In-vitro cultured thymoma tumor cells were positive for SSTR3 only. 

A study that used ^111^In-DTPA-octreotide scintigraphy to explore SSTR expression found that 16 of 17 thymoma cases (94%) were positive for SSTR on scan, while all 4 thymic hyperplasia cases were negative. Based on these observations, octreotide has been anecdotally used with some clinical benefit [25]. 

### 3.7. Merkel Cell Carcinoma (MCC)

Merkel cell carcinoma (MCC) is an aggressive skin cancer where immunotherapy has shown significant activity. However, immunotherapy refractory cases do not have viable treatment options, and mortality is high. Knowledge of somatostatin expression can be utilized for both diagnosis and treatment. A recent case report from Italy documented a metastatic MCC [26] patient who presented with loco-regional relapse after surgery and chemoradiotherapy. Octreoscan and IHC revealed increased expression of somatostatin receptors. The patient was then started on monthly octreotide with good control of the disease over the course of 2 years. Later, the patient was given four cycles of avelumab with continuing octreotide, and remission was achieved. 

Another study [27] analyzed SSTR2A and SSTR5 with IHC in MCCs. Data were collected from a French cohort, and 105 samples were taken from 98 patients. Among these, SSTR expression was noted as follows: SSTR2A (59.2%), SSTR5 (44.9%), and at least one SSTR in 76.5% of tumors. 

Another retrospective study [28] assessed 40 patients with metastatic MCC for SSTR expression by somatostatin scintigraphy. Overall, 85% of cases showed uptake on somatostatin scintigraphy. Somatostatin analogues were used to treat 19 patients, of which 7 demonstrated a response. The rest 12 cases had no response; however, 5 among these 12 showed disease control. 

Papotti et al. documented similar results in their study from 1999. They [29] assessed SSTR2 expression in primary and metastatic Merkel cell carcinoma. The study analyzed 10 cases by RT-PCR and Southern blot. Variable detection of SSTR2 mRNA transcripts and protein expression was detected in 90% of cases. 

In addition, a retrospective analysis from Seattle [30] analyzed 45 cases with metastatic MCC. ^111^In-pentetreotide was used to assess SSTR uptake; 35 patients (78%) showed uptake for SSTR2 and SSTR5. Overall, 33 of these patients were treated with octreotide. Among them, 19 patients had a variable response. The other 12 cases received radiotherapy concurrently with octreotide. Further efficacy analyses [31] were pending at the time of manuscript preparation.

### 3.8. Thoracic Neoplasms

SSTR expression has been studied in lung cancer; however, SSTR expression seems limited in non-small cell lung cancer (NSCLC). In contrast, neuroendocrine tumors in the thorax, including small cell lung cancer, can potentially benefit from targeting somatostatin receptors [31]. 

Strauss et al. [32] studied SSTR2 expression in patients with NSCLC using ^68^Ga-DOTATOC with dynamic PET studies. Dynamic studies with FDG were done as well to compare SSTR expression. This study reported moderate uptake of ^68^Ga-DOTATOC for SSTR2 expression in 7 out of 9 patients. In contrast, none of the 8 metastases samples showed any DOTATOC uptake for SSTR2; however, FDG was positive. The lack of SSTR2 expression in metastases could be due to a loss of gene expression by a variety of mechanisms, as compared to primary NSCLC. Muscarella et al. [33] used ^111^In-DTPA-D-Phe(1)-octreotide and real-time quantitative PCR to compare the radiographic expression and mRNA transcript levels of SSTR2A, 3, and 5 in neuroendocrine lung cancers. A total of 21 neuroendocrine lung cancer patients were enrolled in the study and were compared with 24 healthy donors. A significant increase in uptake was found for SSTR2A and SSTR5 in neuroendocrine lung cancer as compared to the control group.

Another study from Germany [34] studied SSTR expression in small cell lung cancer. The study enrolled 21 patients with extensive-stage SCLC and used ^68^Ga-DOTATATE-PET/CT with PET/CT for assessment. Of the 21 cases, 4 were PET-positive, 6 were intermediate, and 11 were negative. IHC staining was done in 19 cases. SSTR2A was noted in 6 cases (1 with mild, 2 with moderate, and 3 with strong staining) and negative in the remaining samples. In contrast, SSTR5 staining was negative in 16 cases, with mild staining in 1 and moderate staining in 2 cases.

A case series from Italy [35] analyzed SSTR expression types 2A and 3 among 218 lung neuroendocrine tumors, which included metastatic carcinoid (24), atypical carcinoid (73), large cell neuroendocrine carcinoma (60), and 61 surgically resected cases of small cell lung cancer using immunohistochemistry. Among 883 cases of surgically resected NETs of lungs from 1989–2007, these 218 “clinically aggressive cases” were selected. SSTR2A expression was demonstrated as follows: control typical carcinoids (TC) (34%), metastatic TC (71%), atypical carcinoids (51%), large cell NEC (33%), and SCLC (38%). 

Kim et al. [36] studied SSTR expression [36] in refractory small cell lung cancer (SCLC) and advanced pulmonary NETs. In total, 9 patients were selected and underwent ^68^Ga-DOTATATE-PET. Overall, 7 of these 9 were high-grade neuroendocrine carcinomas (6 small cell lung cancer, 1 high-grade neuroendocrine carcinoma), and 2 were cases of atypical carcinoids. Among the 7 high-grade NECs, increased SSTR2 uptake was demonstrated in 5 cases (71.4%), while the scan was negative in the other 2 cases.

### 3.9. Meningioma

Meningiomas account for about one-third of all primary brain and spinal tumors. In 2000, Schulz et al. used somatostatin receptor-specific antibodies [37] to study SSTR expression by IHC in 40 randomly selected cases of meningiomas. Among this cohort, 29 cases (70%) were SSTR2A positive, with strong staining observed in 20 cases. To confirm and correlate the specific SSTR expression, a prospective study was done on 16 cases of surgically resected meningiomas. Similar results were seen, and SSTR2 expression was demonstrated in 12 cases (75%).

Wu et al. [38] did a literature review on SSTR2 expression in numerous studies with meningioma. Various techniques, including IHC, RT-PCR, PET/CT, PET MRI, and SPECT SRS were used in different studies. The literature review concluded that SSTR2 was expressed in almost all cases of meningiomas. Although treatment for meningiomas is typically surgery, SSTR targeting can be considered a viable therapeutic approach for inoperable and metastatic patients and should be validated in prospective clinical trials. 

### 3.10. Pheochromocytoma and Paraganglioma

Pheochromocytomas and paragangliomas are rare catecholamine-secreting neuroendocrine tumors. These tumors express somatostatin receptors, and we have tried to review available data on SSTR expression. Leijon et al. [39] studied 151 tumors, including 127 pheochromocytoma and 24 paragangliomas. The uptake intensity for each receptor subtype was graded as negative, intermediate positive and strongly positive. The uptake was strongly positive for SSTR2(74%) and SSTR3(78%) but negative for SSTR1, SSTR4 and SSTR5. It was also seen that 66% of paragangliomas (15/24) and 54% of pheochromocytomas (69/127) were strongly positive for SSTR2. For SSTR3, strongly positive uptake rates were seen in pheochromocytomas (64%) and paragangliomas (62%). Among metastasized tumors, SSTR2 was strongly positive in 71%, and strong expression of SSTR3 (28.6%) was seen.

Another study from the United Kingdom [40] used immunohistochemical staining and analyzed samples from 77 patients with pheochromocytoma and paraganglioma for SSTRs. Among the receptors, increased uptake was seen for SSTR1, 2, 3 and 5. Among these, SSTR3 was most prevalent and was found in 95% of tumors. This knowledge can be used as a therapeutic target for malignant paragangliomas and pheochromocytomas.

Kaemmerer et al. [41] evaluated SSTR expression by immunohistochemistry in 66 paraffin-embedded tumor samples from 55 patients with histologically confirmed paraganglioma. The receptor prevalence uptake was demonstrated as follows: SST2A (89%), SST5 (47%), SSTR3 (35%), SSTR1 (35%) and SSTR4 (14%). However, no correlation was seen between SSTR expression and grading of tumors or between primary and metastatic tumors.

### 3.11. Head and Neck Squamous Cell Carcinoma 

Head and neck cancers are known to contribute to almost 5% of cancer-related deaths, and prevalence is higher in older males. A study from Japan [42] assessed SST and SSTR1 methylation expression using quantitative RT-PCR. Somatostatin (SST) is an antisecretory and inhibitory hormone that mimics a tumor suppressor gene. Hypermethylation of the SST and SSTR1 promoter gene can lead to gene silencing. Samples were taken from 36 head and neck squamous cell cancers and from 36 normal tissues. The study observed that SST methylation was found in 81% of head and neck squamous cell carcinoma and was seen to correlate with tumor size. In addition, SSTR1 methylation was noted in 64% of samples, and in these, a correlation was observed between tumor size and staging.

Another group of researchers from Cambridge University [43] analyzed somatostatin receptors in 78 head and neck squamous cell carcinoma specimens. The receptor expression was demonstrated as follows: SSTR5 (82%), SSTR1 (69%), and SSTR2 (54%). SSTR3 and SSTR4 expression was rarely observed. 

#### 3.11.1. Laryngeal Squamous Cell Carcinoma

Among head and neck cancers, laryngeal cancers need to be discussed separately. The SEER data from 2015–2019 report an incidence of 2.8 per 100,000 men and women per year and a death rate of almost 0.9 per 100,000 men and women per year. Of these, glottic cancers comprise 2/3rd of cases and typically present earlier with the symptoms of hoarseness. However, supra- and sub-glottic cancers usually present at advanced stages, with metastases leading to increased mortality. A study from China [44] collected samples from 87 laryngeal squamous cell carcinoma patients. Bisulphite pyrosequencing technology was used to measure DNA methylation. Significant associations were noted in patients with different grades and stages of laryngeal carcinoma for SSTR2 methylation as compared to normal tissues. Breakdown analysis later revealed that association was seen mainly in patients <60 years of age and not in patients >60 years. This association was noted in the different staging of laryngeal carcinoma patients. The study also concluded that SSTR2 promoter methylation is a poor prognostic factor for laryngeal squamous cell carcinoma. This emphasizes the need for further studies to expound on the direct correlation of SSTR2 methylation with the grading and staging of laryngeal carcinoma and utilize it for prognosis prediction as well as treatment.

#### 3.11.2. Nasopharyngeal Carcinoma

Similar to supra- and sub-glottic laryngeal cancers, nasopharyngeal cancers may remain asymptomatic for a prolonged period because they originate from the fossa of Rosenmuller, an occult site that does not result in the development of early symptoms. As a result, they usually present at an advanced stage [45]. In 2002, Loh et al. obtained biopsy specimens from 12 nasopharyngeal carcinoma (NPC) patients, and 5 subjects without tumors served as a control group. Tyrosine-labelled I^125^ octreotide and ^125^I-labelled somatostatin 28 were used to study SSTR expression. Moderate to high SSTR2 expression was noted in 9 of 12 patients (75%) with nasopharyngeal carcinoma. Later in 2015, a study from Austria [46] used ^68^Ga-DOTA-TOC-PET/CT to check somatostatin receptor uptake in EBV-positive nasopharyngeal carcinoma. An increased uptake, comparable to well-differentiated NETs, was noted in all five patients in the study. This observation may open new diagnostic and therapeutic opportunities in a subset of patients with NPC.

### 3.12. Breast Cancer

Globally, breast cancer is known to be the most common cause of death in women. The presentation varies from early-stage to locally advanced or metastatic. Since somatostatin is diffusely expressed, its expression in breast cancer specimens could be channelized for diagnostic or therapeutic approaches. In 2005, a group of researchers from Canada [47] used RT-PCR and IHC and reported SSTR subtype expression as follows: SSTR1 (84%), SSTR2 (79%), SSTR3 (89%), SSTR4 (68%), and SSTR5 (68%). Among these, correlation with estrogen receptor (ER) levels was seen with SSTR1, 2, and 4, while a correlation with progesterone receptor (PR) levels was seen only with SSTR2.

In contrast, varying results were documented by Zou et al. in 2019. They studied SSTR expression in 160 ductal breast carcinoma specimens using IHC [48]. Any changes in cell proliferation via somatostatin receptor expression were monitored with flow cytometry. The breast carcinoma specimens included 83 poorly differentiated, 54 moderately differentiated, and 23 well-differentiated ductal carcinomas. The SSTR expression levels were noted as follows: SSTR1 (90%), SSTR4 (71.3%), SSTR5 (44.4%), SSTR3 (41.9%), and SSTR2 (34.4%). The study also mentioned that levels of SSTR expression were inversely correlated with tumor differentiation. No association was noted between receptor expression and hormone receptors (ER/PR). These inferences call for further studies to elaborate on the expression of SSTR and ER/PR hormone receptors.

### 3.13. Ovarian Tumors

Among gynecologic malignancies, ovarian cancers are the second and third most common tumors in developed and developing countries, respectively. In 1991, a group of researchers in Switzerland explored [49] somatostatin receptor expression in 57 specimens of ovarian tumors using in vitro autoradiography with ^125^I-labelled somatostatin ligands. The samples included 51 epithelial tumors, 4 sex-cord stromal tumors and 2 germ cell tumors. Among them, only three were noted to be SSTR positive; all 3 were well-differentiated epithelial tumors. These specimens were also positive for epidermal growth factor receptor (EGFR) but negative for ER and PR. The study reported no somatostatin receptor expression in healthy ovarian tissues.

Later in 2002, a more extensive study was done with specimens from pathology archives at NHS Hospitals Trust, UK [50]. The study included samples from a cohort of benign (35) and malignant (63) ovarian tumors. Of these, 26/35 benign and 57/63 malignant (90%) tumors were positive for somatostatin receptors. Among malignant tumors, SSTR prevalence was seen as follows: SSTR1 (76%), SSTR2 (77%), SSTR3 (29%), and SSTR5 (71%). The study also reported increased levels of SST1 and SST2 expression in the vasculature of benign as well as malignant ovarian tumors. 

Another study [51] analyzed 47 samples via IHC. The receptor prevalence was noted as follows: SSTR1 (19%), SSTR2A (28%), SSTR3 (42%), SSTR4 (17%), and SSTR5 (21%).

More studies are needed to elucidate and categorize SSTR expression in various ovarian tumors, including epithelial, sex cord-stromal and germ cell origin. The expression of SSTR based on vascularity in benign as well as malignant tumors can also give insight into their prognostic indicators.

### 3.14. Malignant and Uveal Melanoma

Malignant melanoma is one of the most aggressive tumors of the skin, and survival rates primarily depend on the stage at which the cancer is diagnosed. Advancements in immunotherapy have improved outcomes in patients with metastatic disease. However, treatment options for immune checkpoint inhibitor refractory cases are limited. SSTR expression was evaluated in melanoma for possible therapeutic approaches in a single small study. In 2001, Lum et al. [52] analyzed 23 specimens from 17 patients with melanomas using ^111^In-pentetreotide octreoscan and RT-PCR. The SSTR expression was noted as follows: SSTR1 (96%), SSTR2 (83%), SSTR3 (61%), SSTR4 (57%), and SSTR5 (9%). However, only 63% of these tumors could be detected by octreoscan, thus creating a huge mismatch. However, SSTR2 expression was noted to transcribe into a functional protein that could bind octreoscan. A thorough analysis of SSTR expression in a larger patient pool may help to come up with novel approaches for targeted therapy.

#### Uveal Melanoma

Among melanomas, uveal melanoma has a very poor prognosis. As uveal melanomas originate from neural crest cells, somatostatin receptor expression in them has also been explored.

A study from Hungary in 2018 used RT-PCR, a ligand-binding assay, and Western blot [53] analysis to study SSTR expression and mRNA transcript levels in 46 specimens of human uveal melanoma. SSTR expression was noted in 70% of specimens. Among the somatostatin receptors, the expression was highest with SSTR2 (65.2%) and SSTR5 (66.6%).

Another study previously evaluated patients with extensive uveal melanoma for somatostatin receptor expression using IHC, scintigraphy, and SPECT [54]. Thirty-one patients were pooled. Among these, 46% displayed SSTR expression via indium-octreotide immunohistochemistry. Among all these cases, staining was positive for SSTR2A. Among these, 7 were given octreotide LAR as a therapeutic intervention. Post-octreotide LAR, 2 died within 1 month after the first dose, while the other 2 had stable disease for almost 5 months. To sum up, as some disease stability was noted with octreotide LAR administration, early introduction of therapy targeting SSTR2A could be studied with long-term effects on prognostic and survival rates.

### 3.15. Neuroblastoma

In [55], 69 cases from a large cohort with variable risk neuroblastoma, including 36 high-risk and 33 non-high-risk tumors, were studied [55]. SSTR2 expression was evaluated using IHC DOTA-Tyr-octreotide. The receptor expression was higher in non-high-risk tumors. In addition, the MIBG avid group was found to have higher SSTR2 expression when compared to the MIBG non-avid group. No correlation between SSTR expression was seen in relation to age, stage, or histological features. SSTR expression was noted to have a significant correlation to the differentiation status. The lower/intermediate-risk neuroblastoma tumors had significant SSTR2 expression compared to high-risk tumors. This could help estimate the disease burden and plan treatment courses accordingly. Even though SSTR2 expression was noted at diagnosis, with worsening disease progression, downregulation was observed. 

## 4. Conclusions

Based on the observations in the above-mentioned studies, somatostatin receptors or relevant mRNA transcripts were noted to be prevalent in extragastric lymphomas, Hodgkin lymphoma, non-Hodgkin lymphoma, DLBCL, gastric cancer (SSTR2), thymoma (SSTR3), Merkel cell carcinoma (SSTR2), some subsets of lung cancer (SSTR2), meningioma (SSTR2), pheochromocytoma/paraganglioma (SSTR2, SSTR3), head and neck cancer (SSTR1, 2, 5) and certain gynecologic cancers (SSTR1). Among benign conditions, SSTR5 has been noted to be prevalent in ectopic and eutopic endometrial tissue, which is often difficult to differentiate from neoplasms. Variable methodologies used for SSTR assessment in source studies are among the limitations of this review. RT-PCR assessment of SSTR mRNA transcript can be hypothesis-generating but is certainly not ideal for bio-selecting patients for therapeutic targeting. Table 1 summarizes the prevalence of somatostatin receptors in various cancers based on the literature review. 

SSTR expression can be potentially used to help assess the burden of disease in some tumors, including extragastric lymphomas, non-Hodgkin’s lymphomas, endometrial neoplasm, gastric cancer, and head and neck cancers. Variable expression was seen in neuroblastoma. SSTR negativity in some cancers is inversely related to prognosis, including prostate cancer.

Overall, there is enough evidence to warrant further evaluation of SSTR expression in certain select tumor types, especially in the era of theranostics where safe and effective SSTR targeting is now feasible with the help of agents like lutetium 177 dotatate. 

## Figures and Tables

**Figure 1 pharmaceutics-14-01394-f001:**
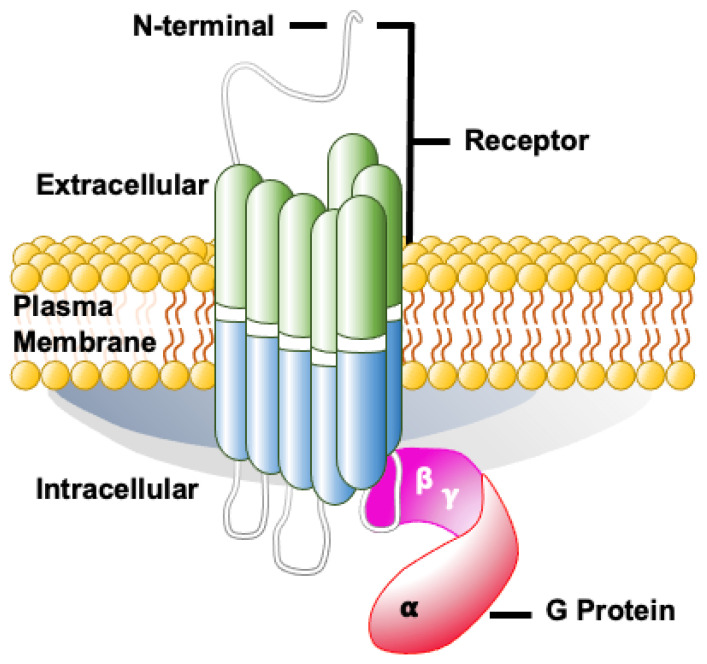
Illustration of somatostatin receptor.

**Figure 2 pharmaceutics-14-01394-f002:**
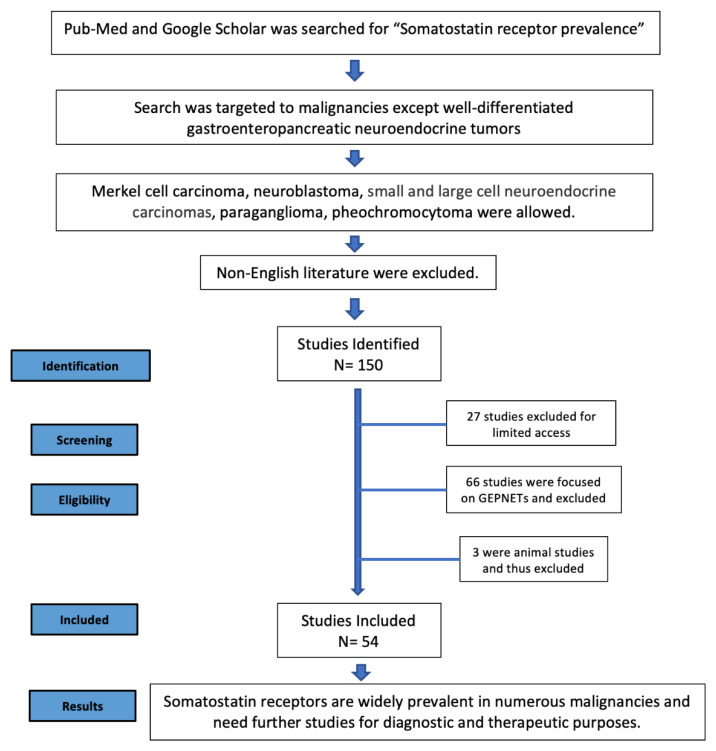
PRISMA flow diagram.

**Table 1 pharmaceutics-14-01394-t001:** Tabular presentation of somatostatin receptor prevalence in various cancers. (The name of the study, number of patients in the study and the analysis are included in parenthesis).

Cancers		SSTR5	SSTR4	SSTR3	SSTR2	SSTR1
MALT-type lymphoma (Stollberg et al., 55, protein) [10]	Gastric and extragastric origin	50%	18%	35%	27%	2%
	Gastric lymphoma	Higher SSTR3, SSTR4, and SSTR5 expression than extragastric tumors.				
	Extragastric tumor	SSTR5 negativity: inversely associated with patient outcome				
Endometrial cancer (Zhao et al., mRNA) [13]	Ectopic endometrium (15)	96.7%	50%	53.3%	70%	43.3%
	Normal endometrium (15)	64.3%	28.6%	21.4%	7.1%	7.1%
Prostate cancer	Werner et al. (276, protein) [14]	10.5%		0.7%	9.1%	0%
	Halmos et al. (80, mRNA) [17]	64%			14%	86%
Gastric cancer	Romiti et al. (51, protein) [18]				SSTR2A 74.5%	
	Hu et al. (80, mRNA and protein) [19]			62.5% (normal gastric mucosa).		
				25% (gastric cancer)		
Merkel cell cancer	Gardair et al. (98, protein) [27]	44.9%			SSTR2A 59.2%	
	Papotti et al. (10, mRNA) [29]				90%	
Thyroid carcinoma and thymoma (Ain et al., mRNA) [23]	Normal thyroid (5)	60	0	100	60	20
	Follicular Adenoma (2)	Present	None	Dominant	Present	Dominant
	Papillary CA (5)	Faint	Faint	Faint	Faint	Faint
	Anaplastic CA (2)	Variable	Very few	Variable	Faint	Variable
	Normal Thymus (3)			Strongly positive	Strongly positive	Strongly positive
	Thymoma (1) (loss of somatostatin production)			Strongly positive		
Lung cancer (Kim et al., 9, uptake) [36]					71.4%	
Meningioma (Schulz et al., 40, protein) [37]					75%	
Pheochromocytoma and paraganglioma	Leijon et al. (151, uptake) [39]	0%	0%	78%	74%	0%
	Parvizi et al. (77, uptake) [40]			95%		
	Kaemmerer et al. (55, uptake) [41]	47%	14%	35%	SST2A 89%	35%
Head and neck cancer	Misawa et al. (36, methylation) [42]					64%
	Schartinger et al. (78, mRNA) [43]	82%	Rare	Rare	54%	69%
Laryngeal cancer (Shen et al., 87, methylation) [44]					Significant association and independent prognostic factor	
Nasopharyngeal cancer (Loh et al., 12, protein) [45]					75%	
Breast cancer	Zou et al. (160, protein) [48]	44.4%	71.3%	41.9%	34.4%	90%
	Kumar et al. (98, mRNA) [47]	68%	68%	89%	79%	84%
Ovarian cancer	Hall et al. (63, protein) [50]	71%		29%	77%	76%
	Schulz et al. (47, protein) [51]	21%	17%	42%	28%	19%
Malignant melanoma (Lum et al., 17, mRNA) [52]		9%	57%	61%	83%	96%

## Data Availability

Not applicable.

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
