# Peer review of "Comprehensive Assessment of Somatostatin Receptors in Various Neoplasms: A Systematic Review"

_pharmaceutics, 2022, doi:10.3390/pharmaceutics14071394_

Round 1
Reviewer 1 Report
The manuscript by Priyadarshini et al. systematically reviews the literature to provide information on the expression of somatostatin receptors (SSTRs) across a wide spectrum of non-neuroendocrine malignancies. Overall, the manuscript is poorly written and lacks conciseness, while the chosen topic is interesting and new. Table 1 is informative, and represents the real message of the work. Figure 1 does not add meaningful contents, and I suggest its removal. References are appropriate. Please see my specific comments below to improve the quality of the manuscript.
Specific comments
1. Authors should carefully re-read their manuscript to check the text for mistakes/consistency. Indeed, sentences like " However, knowledge of receptor expression in various non-neuroendocrine neoplasms is limited to rarer subsets of neuroendocrine neoplasms like merkel cell carcinoma, small and large 13 cell neuroendocrine carcinomas, paraganglioma/pheochromocytoma and neuroblastoma" or "Scans were also used for the effectiveness of therapy for the resolution or persistence of disease" do not make sense to me.
2. Data is the plural of datum. Please edit the verbal forms in the text accordingly.
3. Abstract. "Our manuscript is one of the only such system-19 atic reviews which synthesizes all available data on SSTR expression across tumor types in an easy-to-read format". Please delete this sentence.
4. Please be consistent when referring to DOTATATE. Sometimes Authors use DOTAT-TATE sometimes DOTATATE etc
5. Recently, Lu177DOTA-TATE was 57 approved by FDA for its ability to target small tumors with precision. The FDA approval is not only for small tumors. As it is, this sentence is not correct.
6. If the scope of this manuscript is to summarize the available literature on non-neuroendocrine neoplasms, there is no need to include neuroendocrine neoplasms such as Merkel cell carcinoma, pheo/para and lung NETs/NECs. Moreover, the word Merkel is often mispelled.
7. Line 89. Please change Lymphoma in lymphoma.
8. I am not sure the current manuscript organization is the best one. In particular, one may wonder why the actual results (% of SSTR expression in non-neuroendocrine neoplasms) are reported in the Discussion section rather than in the Results. Authors should globally rethink to the manuscript oranization.
9. PCR (and its technical variants including RT-PCR) is not able to detect levels of expression, but only the levels of the transcript. Please bear this in mind when revising the manuscript.
10. SSTR is sometimes mispelles, i,e. line 119. Sometimes Authors refer to it as "serotonin receptors", that is clearly wrong.
11. Authors should report the technique used for SSTRe expression evaluation when citing ref. 9.
12. Although the Krenning score (Line 130) is well known to physicians or scientists involved in the NET field, not everyone out of the field might be aware of it. Please edit the text accordingly.
13. Line 138. "NHL". Never been abbreviated before.
14. Line 139. "Serotonin scintigraphy". I am not aware of any serotonin scintigraphy.
15. Line 159-160. There is something wrong with numbers here. Please edit.
16. Line 168-169. This sentence can be removed.
17. Line 171. Please change "observed" in "investigated"
18. Line 165-178. Overall, this paragraph is very confused. Please edit.
19. Line 180-181. This sentence does not make any sense.
20. Line 195-196. This sentence does not make sense.
21. Line 223. What correlation are Authors referring to?
22. Line 223-236. This paragraph is not really pertinent to the scope of the manuscript. Please edit.
23. In this section, Authors first discuss colorectal cancer, then gastric cancer, then colorectal again. This is not appropriate and should be edited.
24. Lines 259-261. I do not get the sense of this sentence.
25. Line 266-267. I really do not get this.
26. Lines 281-283. i would erase this sentence.
27. Line 418-419. Promoter hypermethylation leads to reduced expression, not to loss of function.
28. Line 434. There are some problems with the numbers in the incidence.
29. Line 530. Please delete "immense potential".
30. Please change "sandostatin" in "octreotide LAR" whenever it occurs.
31. Line 568-570. There is something missing in this sentence, please edit.
32. Lines 572-574. Please delete this sentence.
33. Table 1 is not clear as it is. Please indicate that percentages (%) are reported, when available. Moreover, be consistent (in some rows the are pure numbers, in other percentages).
34. Line 591-592. Acromegaly and pituitary adenomas have not been discussed in this work. Please remove.
Author Response
Reviewer 1:
- Authors should carefully re-read their manuscript to check the text for mistakes/consistency. Indeed, sentences like " However, knowledge of receptor expression in various non-neuroendocrine neoplasms is limited to rarer subsets of neuroendocrine neoplasms like merkel cell carcinoma, small and large 13 cell neuroendocrine carcinomas, paraganglioma/pheochromocytoma and neuroblastoma" or "Scans were also used for the effectiveness of therapy for the resolution or persistence of disease" do not make sense to me.
Revision: Thank you for a thorough review. We have made necessary revisions and removed/modified the erroneous sentences.
- Data is the plural of datum. Please edit the verbal forms in the text accordingly.
Revision: Thank you. We have reviewed the manuscript for grammatical errors and made necessary changes.
- "Our manuscript is one of the only such system-19 atic reviews which synthesizes all available data on SSTR expression across tumor types in an easy-to-read format". Please delete this sentence.
Revision: We have deleted this sentence.
- Please be consistent when referring to DOTATATE. Sometimes Authors use DOTAT-TATE sometimes DOTATATE etc
Revision: Have addended and used DOTATATE throughout the manuscript.
- Recently, Lu177DOTA-TATE was 57 approved by FDA for its ability to target small tumors with precision. The FDA approval is not only for small tumors. As it is, this sentence is not correct.
Revision: The Lu177 sentence was corrected and modified. Citation [6] added, line 106.
- If the scope of this manuscript is to summarize the available literature on non-neuroendocrine neoplasms, there is no need to include neuroendocrine neoplasms such as Merkel cell carcinoma, pheo/para and lung NETs/NECs. Moreover, the word Merkel is often misspelled.
Revision: We have corrected typographical errors throughout the manuscript. Neuroendocrine neoplasms are a spectrum of disease. Somatostatin receptor expression in well differentiated NETs is well documented, however SSTR expression in various other subsets of neuroendocrine neoplasms are not well known. We would like to include these subsets. We have defined our inclusion criteria in methods section
- Line 89. Please change Lymphoma in lymphoma.
Revision: Changed to “lymphoma” now line 154.
- I am not sure the current manuscript organization is the best one. In particular, one may wonder why the actual results (% of SSTR expression in non-neuroendocrine neoplasms) are reported in the Discussion section rather than in the Results. Authors should globally rethink to the manuscript oranization.
Revision: Given the vast nature of this topic we felt the current organization does justice and provides readers with relevant information on any particular neoplasm they are looking for (with the result and discussion side-by-side). However, we agree with your concern and have renamed bullet 3 to read as “Results and Discussion”.
- PCR (and its technical variants including RT-PCR) is not able to detect levels of expression, but only the levels of the transcript. Please bear this in mind when revising the manuscript.
Revision: Thank you. We have kept this in mind during revision and made appropriate changes.
- SSTR is sometimes mispelles, i,e. line 119. Sometimes Authors refer to it as "serotonin receptors", that is clearly wrong.
Revision: SSTR was corrected in line 184. Serotonin receptors was changed to somatostatin receptors in conclusion. In line 217, serotonin scintigraphy was changed to somatostatin scintigraphy.
- Authors should report the technique used for SSTRe expression evaluation when citing ref. 9.
Revision: The technique “immunohistochemistry” was added to ref 9 (now ref 10).
- Although the Krenning score (Line 130) is well known to physicians or scientists involved in the NET field, not everyone out of the field might be aware of it. Please edit the text accordingly.
Revision: Krenning score was defined in line 207.
- Line 138. "NHL". Never been abbreviated before.
Revision: Abbreviated NHL in line 173.
- Line 139. "Serotonin scintigraphy". I am not aware of any serotonin scintigraphy.
Revision: Changed it to “somatostatin scintigraphy”. Now line 217.
- Line 159-160. There is something wrong with numbers here. Please edit.
Revision: Clarified the numbers. Now 236-237.
- Line 168-169. This sentence can be removed.
Revision: Removed. Now line 243
- Line 171. Please change "observed" in "investigated"
Revision: Sentence modified. Now line 244
- Line 165-178. Overall, this paragraph is very confused. Please edit.
Revision: We have made revisions to the paragraph
- Line 180-181. This sentence does not make any sense.
Revision: This sentence was modified. Now line 310.
- Line 195-196. This sentence does not make sense.
Revision: Line modified. Now 325.
- Line 223. What correlation are Authors referring to?
Revision: The line was modified (now line 373) and was meant to express the correlation mentioned in previous paragraphs.
- Line 223-236. This paragraph is not really pertinent to the scope of the manuscript. Please edit.
Revision: We do understand that this paragraph cites animal studies but this is a very vital piece of information for EBV driven gastric cancers which has not been well studied in human studies yet. We want to give some food for thought to the readers.
- In this section, Authors first discuss colorectal cancer, then gastric cancer, then colorectal again. This is not appropriate and should be edited.
Revision: Changed the sequence of paragraphs. Line 333-386.
- Lines 259-261. I do not get the sense of this sentence.
Revision: We have made necessary edits. Line 425-427.
- Line 266-267. I really do not get this.
Revision: The line was modified. Now line 432-434.
- Lines 281-283. i would erase this sentence.
Revision: We have deleted this. Line 443.
- Line 418-419. Promoter hypermethylation leads to reduced expression, not to loss of function.
Revision: Line modified. (Line 648)
- Line 434. There are some problems with the numbers in the incidence.
Revision: We have corrected the error. (Line 663-665)
- Line 530. Please delete "immense potential".
Revision: Deleted “immense potential” in line 777.
- Please change "sandostatin" in "octreotide LAR" whenever it occurs.
Revision: “Sandostatin” was changed to “octreotide LAR” throughout the manuscript.
- Line 568-570. There is something missing in this sentence, please edit.
Revision: Edited. Line 815-817.
- Lines 572-574. Please delete this sentence.
Revision: We have deleted this. Line 826.
- Table 1 is not clear as it is. Please indicate that percentages (%) are reported, when available. Moreover, be consistent (in some rows the are pure numbers, in other percentages).
Revision: % was added in the table as appropriate. Will keep figure 1 to explain the structure of the 14 amino acid somatostatin receptor.
- Line 591-592. Acromegaly and pituitary adenomas have not been discussed in this work. Please remove.
Revision: Acromegaly and pituitary adenoma was removed from line 858.

Reviewer 2 Report
This article entitled with "Comprehensive Assessment of Somatostatin Receptors in Various Neoplasms” is a systematic review about SSTR expression in various neoplasms except neuroendocrine neoplasms.
There is a minor point to be revised in this manuscript as below.
<Major comments>
1) Nothing.
<Minor comments>
1) I don’t think that endometriosis is included in neoplasm. What do you think? Benign diseases known to be positive on SSTR scintigraphy include ectopic endometriosis, sarcoidosis, and intrapancreatic accessary spleen, which are often problematic to differentiate from neoplasms. These should be described in the end of discussion.
Author Response
Reviewer 2:
- I don’t think that endometriosis is included in neoplasm. What do you think? Benign diseases known to be positive on SSTR scintigraphy include ectopic endometriosis, sarcoidosis, and intrapancreatic accessary spleen, which are often problematic to differentiate from neoplasms. These should be described in the end of discussion.
Revision: We agree with your assessment. SSTR5 prevalence has been noted in ectopic endometrial tissue. We have now described this in discussion. (Line 850-851).

Reviewer 3 Report
Authors review the literature on the expression and activity of somatostatin receptors in several types of cancers.
the manuscript is well written and informative, but some modifications are needed to improve clarity.
It seems that only papers that take into consideration all SSTR. What is the rationale? Thanks to a brief literature search, it seems that SSTR2A has an important role in gliomas.
please include in the table the following informations:
1. number of patients in the studies
2. how SSTRs were analyzed (mRNA, protein, uptake, methylation).
To this regard, a comment on the possible limitations of the different types of analysis. Is mRNA sufficient to hypothesize therapy?
Author Response
Reviewer 3:
- It seems that only papers that take into consideration all SSTR. What is the rationale?
Revision: Thank you. We tried to include any study which described SSTR expression regardless of SSTR subtype. Since data is limited, we were broad in our search.
- please include in the table the following informations: i) number of patients in the studies ii) how SSTRs were analyzed (mRNA, protein, uptake, methylation).
Revision: The table was addended to include number of patients in each study as well as the methods used to analyze SSTR (mRNA, protein, uptake, methylation).
- To this regard, a comment on the possible limitations of the different types of analysis. Is mRNA sufficient to hypothesize therapy?
Revision: We agree with your assessment. mRNA assessment is not ideal but is hypothesis generating. Therapeutic targeting will definitely require imaging-based confirmation of SSTR especially with latest generation gallium 68 or copper 64 dotatate. We have added a comment in conclusion section.

Round 2
Reviewer 3 Report
authors have addressed all points